# A Review on Image Sensor Communication and Its Applications to Vehicles

**Ruiyi Huang** [1,*] and **Takaya Yamazato** [2]

1 Department of Information and Communication Engineering, Nagoya University, Nagoya 464-8603, Japan
2 Institute of Liberal Arts and Sciences, Nagoya University, Nagoya 464-8603, Japan
* Correspondence: huang.ruiyi.c6@s.mail.nagoya-u.ac.jp

**Abstract:** Image sensor communication (ISC), also known as optical camera communication, is a form of visible light communication that utilizes image sensors rather than a single photodiode, for data reception. ISC offers spatial separation properties and robustness to ambient noise, making it suitable for outdoor applications such as intelligent transportation systems (ITSs). This review analyzes the research trends in ISC, specifically concerning its application in ITSs. Our focus is on various ISC receivers, including rolling shutter cameras, global shutter high-speed cameras, optical communication image sensors, and event cameras. We analyze how each of these receivers is being utilized in ISC vehicular applications. In addition, we highlight the use of ISC in range estimation techniques and the ability to achieve simultaneous communication and range estimation. By examining these topics, we aim to provide a comprehensive overview of the role of ISC technology in ITSs and its potential for future development.

**Keywords:** image sensor communication; optical camera communication; intelligent transportation system; visible light communication; rolling shutter camera; high-speed camera; visible light positioning

## 1. Introduction

Visible light communication (VLC) is a wireless communication method that utilizes visible light for information transmission [1–4]. The luminance of the VLC light sources can be modulated extremely fast to make the switching of light imperceptible to the human eye to achieve high-speed data transmission [4]. In contrast to traditional wireless communications that use electromagnetic waves, wireless communication using visible light has a much wider wavelength range and has been considered a technique to address the issues caused by the growing spectrum need for wireless communication [5].

The origins of VLC can be traced back to the 7th century BC, when people in ancient China used smoke and fire to transmit information about their enemies across long distances. Over two millennia later, in the 1880s, Alexander Graham Bell made his groundbreaking attempt to use machinery to implement VLC by transmitting voice information through sunlight, carried by electricity [6]. Unfortunately, however, this technology did not reach early commercialization due to its limitations in communication distance and the inevitable effects of weather and obstacles such as rain. It was not until the 1990s that the invention of a high-brightness light-emitting diode (LED) [7] revolutionized the lighting industry, opening up new possibilities for VLC. The high response and fast switching speed of LEDs allow using visible light to convey high-speed data [4,8]. In 2004, T. Komine and M. Nagakawa proposed an indoor VLC system using white LED lights for both illumination and optical wireless communication [4]. They considered the interference and reflection of the multiple light sources installed in a room and demonstrated its potential for high-speed data transmission of around 10 Gbps. This paper has been highly influential in the field of VLC, leading to widespread research interest and study in VLC. VLC technology is also applied outdoors, for instance, in traffic lights and streetlights. In 1999, G. Pang et al. showed

the potential of using a traffic light as a communication device [9]. In [9], high-brightness LED lights were modulated and encoded with audio messages. The receiver was designed to demodulate the optically transmitted audio messages using a photodiode (PD) to convert the light to electrical signals through direct detection. However, there were many constraints to using a PD in outdoor environments due to the strong ambient noise. In 2001, an image sensor was introduced as a new type of receiver for VLC, which was more robust to outdoor ambient noises [10]. An image sensor consists of an enormous array of PDs. This structure of an image sensor gives it the ability to spatially separate the light sources, also resulting in being robust to outdoor ambient noise and being able to easily track moving vehicles. Nowadays, we have coined the term image sensor communication (ISC) for the VLC that uses image sensors as receivers [11–13]. ISC can also be referred to as optical camera communication (OCC) [14,15]. The primary distinction between ISC and OCC is that the term "camera" in OCC encompasses both the lens and circuitry for an image sensor. Additionally, these two terms are named by different institutions. In this review, we will discuss ISC and its outdoor applications, such as intelligent transportation systems (ITSs), which are also known as ITS-ISC.

For ITSs, low latency and accurate localization are crucial factors [16–18]. Low latency is necessary for real-time communication between vehicles and their surrounding transportation elements to efficiently coordinate traffic flow and avoid accidents timely. Accurate localization is important for car navigation and collision avoidance, as well as for providing location-based services to passengers. ISC can be an applicable technology to realize low latency and accurate localization simultaneously. On the one hand, ISC can achieve low latency by exploiting the high-speed capabilities of digital image sensors, for instance, using high-speed cameras as the ISC receiver [19] or utilizing the rolling shutter effect [20]. The high-speed transmission allows real-time data transfer between vehicles, road infrastructure, and other elements in the ITS. On the other hand, ISC systems can achieve high localization accuracy by exploiting the high-brightness properties of visible light. Even in complex traffic environments, such as tunnels or intersections, LED transmitters can be recognized from complicated backgrounds using algorithms in ISC. In contrast, localization using conventional wireless technologies can be inaccurate due to signal reflection, multi-path propagation, or environmental interference [21]. Furthermore, many existing automotive applications are now equipped with image sensors, such as driver recorders and autonomous driving sensors [22]. Meanwhile, the road and vehicles are equipped with LED lights. Therefore, ISC that utilizes both image sensors and LEDs can be a promising technology for ITS.

We will provide detailed discussions of the ISC studies in the remainder of this review paper. In Section 2, we present the research trend of VLC based on the statistical data collected on Scopus. We analyze two receiver types of VLC: PDs and image sensors. Then, we discuss the advantages of using image sensors in outdoor environments. It then leads to the research status of ITS-ISC. We also analyze the constitutions of transmitter and receiver types of ISC. In Section 3, we explain the concepts necessary for ITS-ISC, including vehicle-to-vehicle (V2V), vehicle-to-infrastructure (V2I), infrastructure-to-vehicle (I2V), and vehicle-to-everything (V2X) communications. We then describe the architecture of the ITS-ISC system. In addition, the advantages and limitations of image sensor receivers are discussed. In Section 4, a comprehensive review of the receiver types of ISC is provided, including rolling shutter cameras, global shutter high-speed cameras, optical communication image sensors (OCIs), and event cameras (dynamic vision sensors). In Section 5, we illustrate the basic mechanism, challenges, and solutions of range estimation using LED transmitters and image sensor receivers. Finally, Section 6 presents the conclusion.

## 2. Research Trend

### 2.1. Research Trend of VLC

Figure 1a shows the annual publication numbers of VLC literature published before 2023 searched in Scopus. We found a total of 9345 publications. From Figure 1a, it can be seen that the number of VLC publications has greatly increased since around 2010 but declined after 2019. Figure 1b shows the results of classifying these research publications by document type. According to Figure 1b, conference papers constitute 50.3% of the total. This might be the reason behind the decline in publication numbers from 2019 when the COVID-19 pandemic started making it challenging for people to attend international conferences.

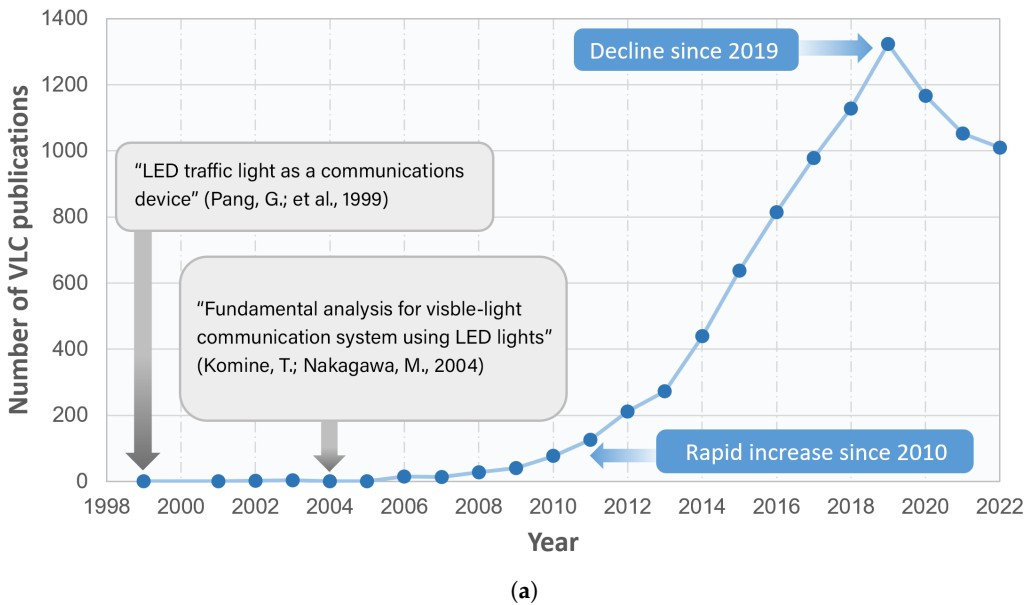

**(a)**

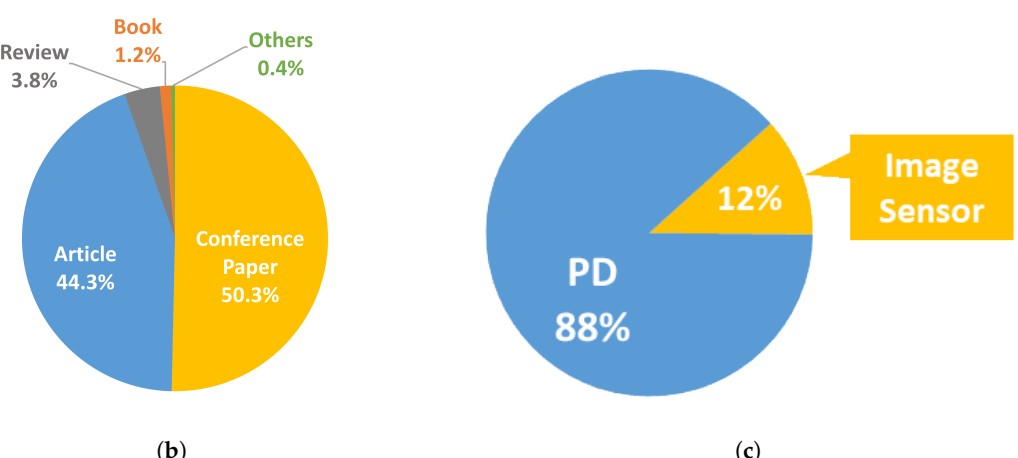

**(b)**                                                                 **(c)**

**Figure 1.** Literaturesurveys on visible light communications (VLC). The survey was performed by searching the Scopus database for documents with "visible light communication" in the title, keywords, and abstracts, and publication year before 2023. These data are based on search records in February 2023. (**a**) Annual growth in the number of research publications related to VLC [4,9]. (**b**) Analysis of document types in research publications of VLC. (**c**) Categorizing receiver types in research publications of VLC.

Figure 1c describes the percentage of publications using PD or image sensor receivers in VLC. 88% of the publications used PDs, and the number was 8194 publications. Only 12% of the publications used image sensors, with 1151 publications. Therefore, it can be

concluded that most works utilize PDs as the receiver. There are many advantages to using PDs, such as rapid response and low cost. Due to the high-speed response, the method of using a PD as a receiver device has achieved communication at the Gbps level [5,23,24]. In contrast, image sensors communicate much slower than PDs due to the frame rate limitation. Based on our survey, research on PDs is being conducted mainly for indoor use purposes. For example, illumination optical communication including Light-Fidelity (LiFi) [5] and indoor visible light positioning [25]. In terms of image sensor research, its proportion in indoor use is lower compared to the research on PDs.

Image sensors are more suitable for outdoor applications than PDs. The main reasons are that image sensors are more robust to solar noise than PDs and are simpler to track moving vehicles. When used outdoors, especially in an ITS environment, the effect of background light noise, such as sunlight and other light sources, cannot be ignored. Therefore, for a PD, the viewing angle must be narrowed to ensure an adequate signal-to-noise ratio. At the same time, mechanical manipulation is required to orient the PD toward the direction of the transmitter. However, when the car is driving at high speed and changing its direction, it is difficult for a PD with a narrowed viewing angle to swiftly adjust the direction of its optical axis. On the contrary, an image sensor can track moving vehicles with little adjustments to its direction since it has a large number of pixels. Moreover, an image sensor is capable of spatially separating light sources and selecting only the pixels that contain the desired transmitted signal while filtering out other pixels that may contain background light noise.

Additionally, there are ongoing efforts to establish international standards for VLC. In 2011, the IEEE 802.15.7 standard for short-range wireless optical communication was established [26]. It was revised later to include ISC and new PHY modes in IEEE 802.15.7a task group [13]. In July 2020, a new amendment to IEEE 802.15.7a was proposed, which adds a deep learning mechanism for ISC. This amendment is expected to be published in September 2023. Furthermore, in December 2017, a proposal for a new ISO standard for localized communication using ISC in ITSs was approved. The proposal suggested a new communication interface called "ITS-OCC", which adopted ISC profile, communication adaptation layer, and management adaptation entity from IEEE 802.15.7:2018, ISO 14296:2016, and ISO 22738:2020, respectively [27–29]. The ISO 22738:2020 standard was published in July 2020 and is expected to facilitate the development of OCC-based ITS applications [29]. Finally, regarding the performance comparison between IEEE 802.15.7 for optical wireless communication and IEEE 802.15.7a for OCC, the addition of image sensor receivers and deep learning mechanisms in IEEE 802.15.7a OCC is expected to improve the accuracy and robustness of communication performance and range estimation in challenging environments. It may also allow for the development of new applications that were previously not possible with traditional range estimation methods. Further research and development in this area may lead to improved performance and new use cases for ISC.

### 2.2. Research Status of ISC

Figure 2 shows the growth in the number of ISC literature published before 2023 searched in Scopus. We found a total of 1151 papers. There is a significant increase in the quantity of literature starting in 2013; the growth in the number of papers levels off after 2019. Roughly 24% of this literature is applied to vehicles.

We selected papers with the literature type of "article" and "conference paper", and classified these publications by applications, transmitter types, and receiver types. Note that the publications were categorized manually by the authors; subjective interpretation and potential errors could be introduced into the categorization process.

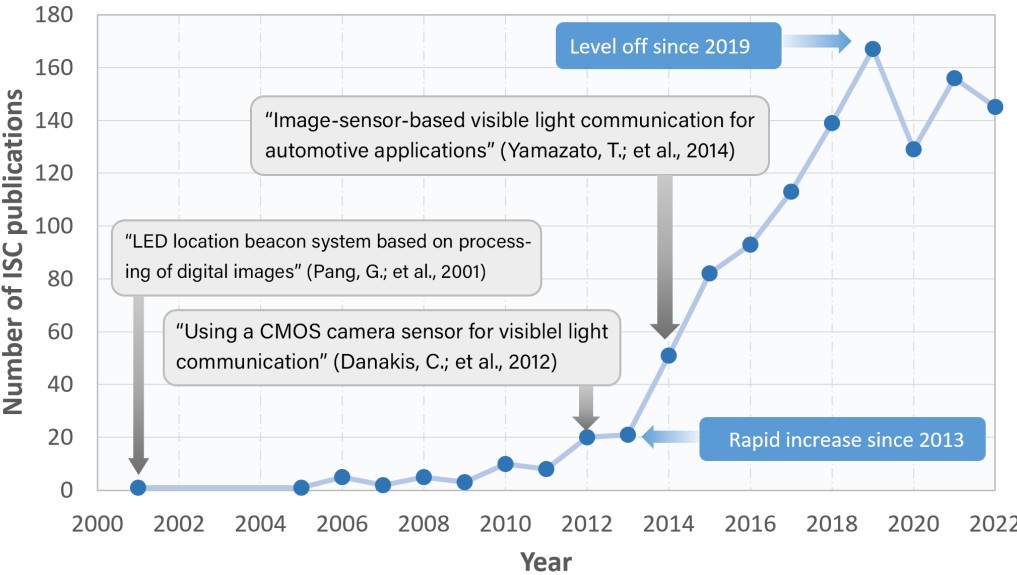

**Figure 2.** A literature survey of image sensor communication (ISC) [10,19,20]. The survey was performed by searching the Scopus database for publications with "image sensor communication" or "optical camera communication" or "screen camera communication" or "display camera communication" or {"visible light communication" and "camera"} or {"visible light communication" and "image sensor"} in the title, keywords, and abstracts, and publication year before 2023. These data are based on search records in February 2023.

Figure 3 shows the proportions of publication numbers of different ISC applications. Communication application accounts for 78.7% of the whole. Here, communication refers to being able to transmit information such as text, image, voice, and video through optical signals. Papers with communication applications cover a wide range of subjects, such as bit-error-rate (BER) measurement, communication model design, and other related areas. In the context of an ITS, communication enables safety and efficiency applications such as collision avoidance, traffic flow optimization, and emergency vehicle notifications. Ranging, ranging application, also considered as localization or positioning, accounts for 13.9% of the whole. Ranging relies on computer vision techniques and can be used for applications such as obstacle avoidance and navigation. By using ISC, high-accuracy positioning can be achieved within two milliseconds, which is faster than light detection and ranging (LiDAR) or global positioning system (GPS) [30]. Additionally, detection and tracking comprise 3.6% of all ISC papers. It refers to extracting LEDs from the received image streams. Inaccurate detection or tracking can lead to incorrect data decoding because it is conducted prior to communication and ranging. The advantage of ISC-based communication and ranging technology is that the use of LEDs provides communication and navigation services along with illumination, thus minimizing the need for additional power. It reduces front-end costs, eliminating the need for additional installation and configuration of signal access points, and reducing the cost and complexity of the communication and navigation system.

Figure 4 shows the ratio of the different transmitters and receivers in the ISC. However, Figure 4b may have some level of inaccuracy. This is because some authors did not specify the type of image sensor used in their articles, and these kinds of articles are classified as "others". Additionally, due to time constraints, the "others" category was not fully categorized. We conclude that since the rolling shutter camera is the majority of the receivers, some instances of rolling shutter cameras may have been categorized as "others".

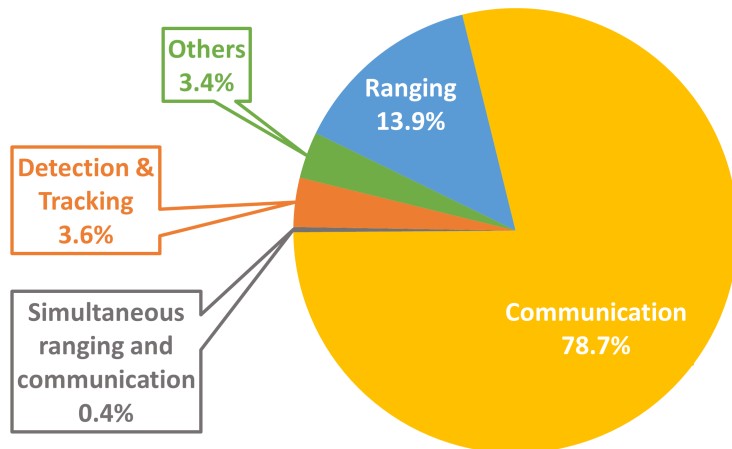

**Figure 3.** Categorizing image sensor communication research publications by applications of communication, ranging, simultaneous ranging and communication, detection and tracking, or others. Others include channel characterizing [31], vehicle vibration modeling [32], etc.

As shown in Figure 4a, the transmitters can be divided into more than two types, including LEDs and screens. It can be seen that LEDs are used in a much higher proportion than screens. LEDs are also the most suitable transmitters for ITS applications. Not only are existing traffic lights equipped with LED arrays, but their switching speed is also the fastest. LED transmitters can also be divided into different categories, for example, single LED, LED arrays, and rotating LED arrays. The number and size of LEDs in the transmitter determine to some extent how much data can be transmitted over how many distances. Because the number of pixels taken up by the transmitter decreases as the distance between the transmitter and the image sensor increases. With regard to the ISC receiver, the receivers can be divided into more than four types as shown in Figure 4b, including rolling shutter cameras, high-speed cameras (global shutter), event cameras, and OCI. They will be discussed in Section 4.

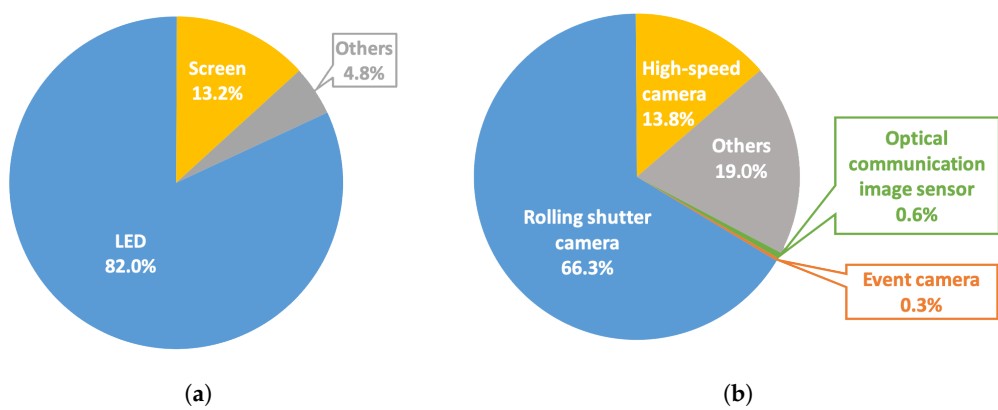

(**a**)                                   (**b**)

**Figure 4.** Categorizing image sensor communication (ISC) research publications by transmitter types and receiver types. The number of publications is 1021. (**a**) Categorizing ISC publications by transmitter type. Others includes micro-LED [33], organic-LED [34], projector [35], laser diodes, etc. (**b**) Categorizing ISC publications by receiver type. Others includes charge-coupled device (CCD) camera, time-of-flight sensor [36], lensless-camera [37], etc.

### 2.3. Image-Sensor-Communication-Based Intelligent Transportation System (ITS-ISC)

Figure 5 shows the proportions of different application scenarios in ISC, where outdoor applications comprise 24% and the number is 251 papers. Despite having a lower proportion than indoor applications, the potential of ISC outdoor applications in the research field of ITSs cannot be overlooked.

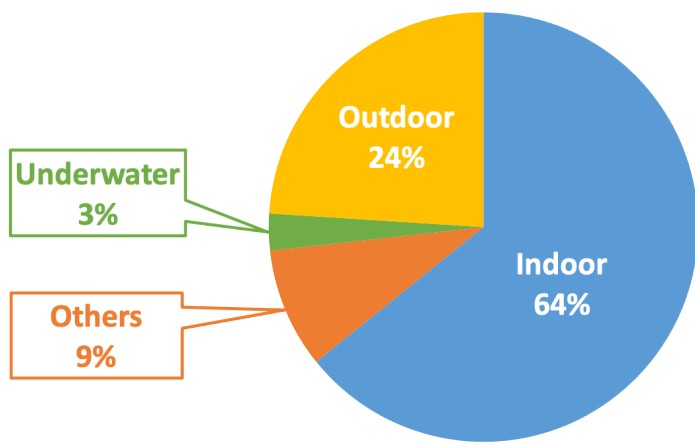

**Figure 5.** Categorizing image sensor communication research publications by application scenarios including indoor, outdoor, underwater, or others.

ITS refers to a system that connects people, roads, and vehicles through the use of communication technologies, aiming to improve road traffic safety and smooth traffic flow by reducing congestion, and achieving comfort dedicated short-range communications (DSRC) [16,38]. A familiar example of an ITS is the electronic toll collection (ETC) system. The latest report shows that the global intelligent transportation market reached a size of 110.53 billion USD in 2022 and is projected to increase at a compound annual growth rate of 13.0% during the forecast period from 2023 to 2030 [39]. This growth trend in the market highlights the need for vehicles to use intelligent transportation services.

The technological development of autonomous vehicles is currently attracting worldwide attention, especially in the field of ITSs. The recognition of the surrounding road conditions using various onboard sensors is important for driving safety, as well as driving support for autonomous vehicles. Cameras are one of the most commonly used onboard sensors in self-driving cars and advanced driver assistance systems [22,40]. These onboard cameras are mainly used for applications such as visual aids, object detection, and driving recording. If these cameras are used as receivers for ISC, they can receive the optical signal from LEDs on the road to provide communication and localization services. Information sent from the vehicle is expected to include location information such as the vehicle's longitude and latitude, vehicle signals indicating driving conditions such as speed, steering, acceleration, braking, or vehicle front view, and vehicle-specific information such as vehicle model, width, and height. Information sent from the side of the road is expected to include traffic signal statuses, such as color and transition time, and traffic information, such as weather conditions and nearby traffic congestion. These kinds of information transmitted through the ISC system can provide efficient and reliable communication for a variety of applications, including traffic management, road safety, and autonomous driving.

In addition, image sensors can help to minimize or manage latency in vehicular networks, which is crucial for safety-critical applications, such as collision avoidance and cooperative driving. ISC can offer better spatial and directional accuracy than conventional radio frequency-based communications, which can enhance the reliability of achieving low latency. Hence, the integration of ISC in vehicular networks can considerably enhance the performance and safety of upcoming ITSs.

## 3. Basic Concept and Architecture of an ITS-ISC

### 3.1. Vehicle-to-Everything (V2X) Communications Using Image Sensors and LEDs

ISC can be implemented in a variety of contexts in an ITS, including V2V, V2I, and I2V communications. These means of communication are involved in vehicle-to-everything (V2X) communications, which refers to any type of communication between a vehicle and its surrounding traffic environment, as illustrated in Figure 6.

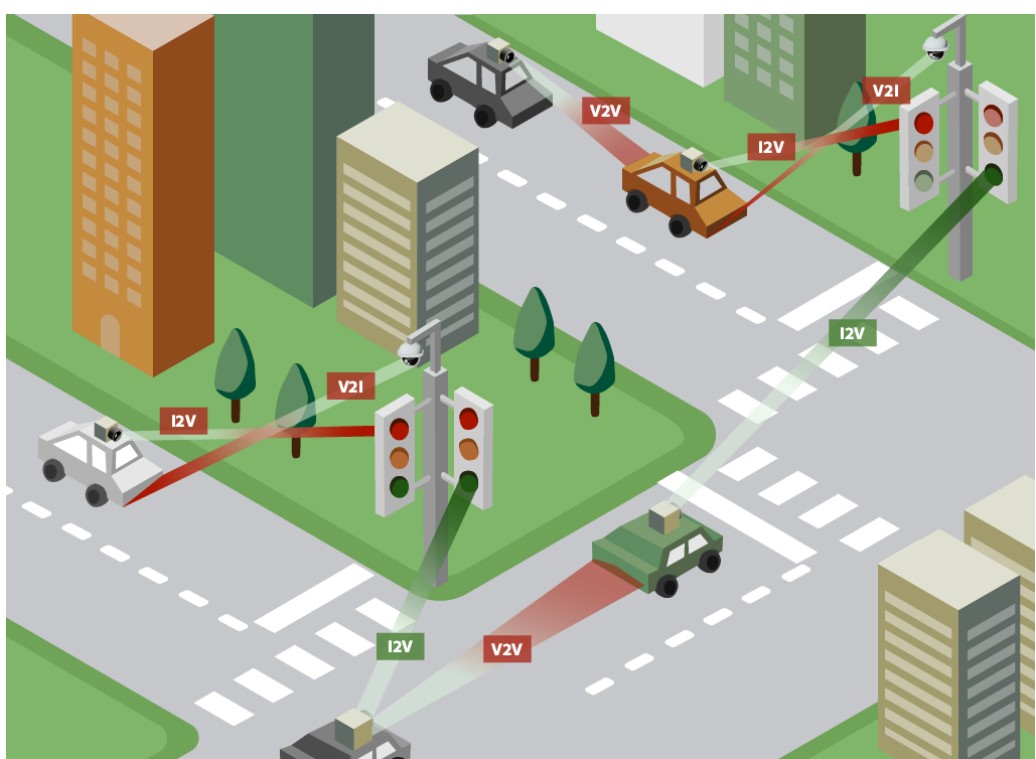

**Figure 6.** The basic idea of image sensor communication for vehicular applications including vehicle-to-vehicle (V2V), vehicle-to-infrastructure (V2I), and infrastructure-to-vehicle (I2V) communications. Data signals are emitted from traffic lights and vehicle headlights and taillights. Image sensors are mounted on the vehicle to receive visible light signals. The image sensor converts the received visible light signal into an electrical signal and decodes the signal using image processing technology.

In V2X communication, vehicles can communicate with each other and share information about their speed, location, and other important data, which helps improve safety and traffic flow. For example, suppose a vehicle detects an obstacle on the road ahead. In that case, the vehicle can transmit this information to the rear vehicles to alert them and allow them to take appropriate action. This type of communication between vehicles is referred to as V2V communication. Moreover, V2X communications enable vehicles to send and receive information about the road environment, such as traffic conditions or road hazards, to improve safety and efficiency. For example, a traffic signal can transmit data to a vehicle about the time remaining before the signal changes, allowing the vehicle to adjust its speed accordingly. Furthermore, the information transmitted via V2X communications may include information about traffic patterns, weather conditions, and road closures. For example, sensors embedded in the road can detect the presence of ice or snow and transmit this information to vehicles to help them navigate safely. These types of communication between infrastructures and vehicles are referred to as I2V and V2I.

However, ISC is restricted to light-of-sight (LOS) situations due to the reliance on visible light. If an obstacle interrupts the light, it will interrupt the information transmission, which is likely to be the case in practical transportation applications such as lane changing. There are solutions to handle such non-light-of-sight (NLOS) situations based on reflections [41]. Alternatively, the vehicle can receive information indirectly through the

obstacle if it is an ISC transmitter. In addition, DSRC can be used for NLOS situations [16]. It transmits by wireless radio waves and can penetrate obstacles. However, DSRC is limited by high levels of noise and multipath effects, which ISC can mitigate. Therefore, in certain situations, a hybrid approach that considers both methods may be appropriate.

### 3.2. Optical Channel Characteristic and Modulation Schemes

The optical channel can be expressed by

$$R(t) = I(t)X \otimes h(t) + A(t) \tag{1}$$

where $I(t)$ is the transmitted waveform, $R(t)$ is the received waveform, $h(t)$ is an impulse response, $A(t)$ represents the noise, $\otimes$ symbol denotes convolution, and $X$ is the detector responsivity. In addition, the average transmitted optical power $P_t$ in the time period $2T$ is given by

$$P_t = \frac{1}{2T} \int_{-T}^{T} I(t)dt \tag{2}$$

The average received optical power $P$ can be presented by

$$P = H(0)P_t \tag{3}$$

where $H(0)$ is the channel DC gain, given by $H(0) = \int_{-\infty}^{\infty} h(t)dt$. In addition, the digital signal-to-noise ratio (*SNR*) is given by

$$SNR = \frac{X^2 P^2}{X_b N_0} \tag{4}$$

where $X_b$ is the bit rate, and $N_0$ is the double-sided power-spectral density [42].

There are several modulation schemes available for ISC, each of which can affect the optical power $P_t$. The most commonly used modulation scheme is on-off keying (OOK), although others such as pulse-width modulation (PWM), pulse-position modulation (PPM), pulse-amplitude modulation (PAM), and orthogonal frequency division multiplexing (OFDM) are also utilized.

### 3.3. Advantages and Limitations in Image Sensor Receivers

An important feature of the image sensor is spatial separation. In ISC, spatial separation refers to spatially separating signals coming from different transmitters, such as traffic signals, street lights, and brake lights of the lead vehicle. A visual explanation of spatial separation is shown in Figure 7.

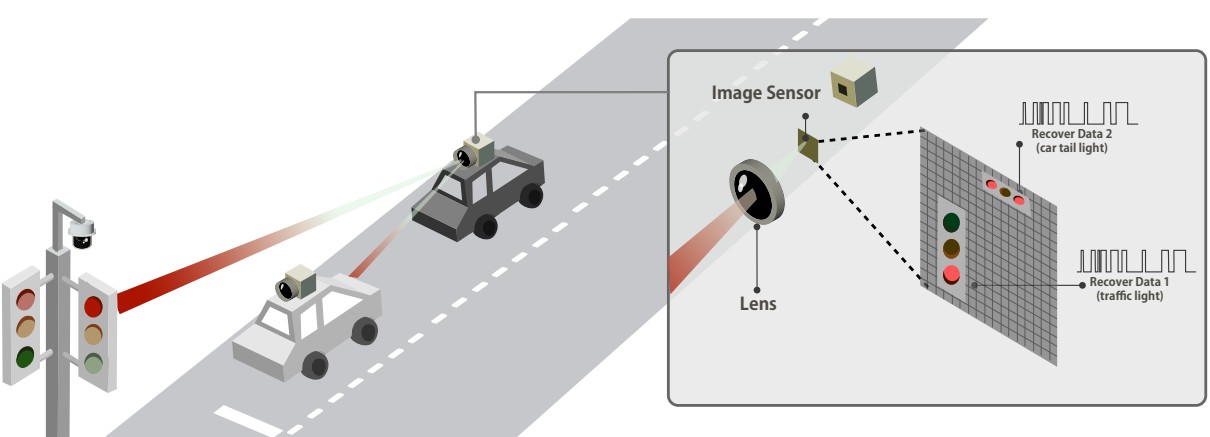

**Figure 7.** A visual explanation of spatial separation. Two cars are driving on the road and the camera of the following vehicle receives two light streams from the traffic light and the backlight of the lead vehicle. An image sensor receiver can separate these two light sources spatially.

Spatial separation enables ISC to perform parallel transmissions using multiple light sources and separating them by image processing. The interference between each light source is significantly reduced by spatial separation, while it is hard to separate the light sources using a single PD due to the interference. An LED array, consisting of multiple LED elements arranged in a grid pattern, can be used to achieve spatial separation [43,44]. Each element in the LED array can be individually controlled to emit light at different intensities. By modulating the light emitted by each LED element, we can encode data into the light signal and transmit it to a receiver. This approach allows for parallel transmissions and can increase the overall data transmission rate. By using an image sensor receiver, we can reduce the impact of interference between light streams from different LEDs. This is particularly useful when dealing with high-speed data transmission, where even small amounts of interference can lead to significant errors. Furthermore, the spatial separation of the image sensor is particularly beneficial in distance measurement. An image sensor captures a wide field of view, allowing obtaining depth information through monocular vision or stereo vision. This can help vehicles make more accurate and safe driving decisions.

However, the effectiveness of spatial separation depends on three factors of the image sensor: resolution, frame rate, and dynamic range. First, regarding sensor resolution, using a low-resolution image sensor will lead to lower data rates and increased vulnerability to errors. A low-resolution image sensor captures images with few pixels, so the amount of information that can be transmitted through ISC is limited to its pixel amount. In addition, if the resolution of the image sensor is low, the interference of the optical signal between adjacent pixels can be significant and can affect the accuracy of demodulation. The low-resolution issue may be solved by image processing algorithms such as Kalman filter and resolution compensation [45]. Second, in terms of frame rate, low frame rates result in low data rates. A low frame rate commonly occurs if an image sensor has a large number of pixels, because a large number of pixels requires high processing ability, resulting in a slow processing speed for each frame. The frame rate of a conventional complementary metal–oxide–semiconductor (CMOS) image sensor is usually around 30 frames per second (fps) [46]. While some commercial cameras or industrial cameras can exceed 30 fps, such as iPhone 14, which has slo-mo video support at 120 fps or 240 fps [47], they sacrifice the video quality. Finally, another problem with ISC is the limited dynamic range of the image sensors. When light irradiates onto the image sensor, the pixels absorb photons and convert them into electrons. These charges then accumulate in the pixel potential well, and when the accumulation limit is reached, no more photoelectric conversion can take place, even if the light is brighter (more photons). In other words, the output digital signal reaches saturation. A low dynamic range leads to a low saturation threshold. If the pixels that contain LEDs are saturated, the demodulation error will be larger, especially with luminance-based modulation such as pulse-width modulation. To solve the issue caused by saturation, we can use a pre-coding method [48] or image processing algorithms, such as phase-only correlation (POC) [49].

## 4. Isc Receivers

To be used effectively in a moving vehicle environment, the ISC signal must be transmitted at high speed. While CMOS image sensors are the most commonly used sensors for ISC, conventional CMOS sensors often have low frame rates that make it challenging to transmit signals at high speeds. As a result, the ISC receivers, namely the image sensors, are required to utilize encoding techniques and image processing algorithms to achieve high-speed transmission. This section analyzes various ISC receivers, including rolling shutter cameras, high-speed cameras, OCI, and event cameras.

### 4.1. Rolling-Shutter Camera

According to our survey in Section 2.2, rolling shutter cameras are the most widely used receivers in ISC. The reasons are that rolling shutter cameras are cheap and are utilized

in most commercial cameras, including dashcams, closed-circuit television, and those on mobile phones.

In the rolling shutter mechanism, each row of the image sensor is exposed sequentially. This feature enables high-speed data transmission in ISC by allocating data to each row of the rolling shutter sensor [20,50]. Figure 8 provides detailed descriptions of the rolling shutter camera. Figure 8a shows a conventional four-transistor rolling shutter circuit. The row scanner selects the rows from top to bottom individually, and then the column scanner controls reading out the data of an entire row. A pixel in the rolling shutter sensor has no ability to store the accumulated charges due to the lack of memory nodes. Therefore, the exposure process should be followed by an immediate readout to ensure consistent exposure time for each row. This results in the rolling shutter mechanism shown in Figure 8b. The row pixels are read out sequentially thus the reset time starts sequentially. This rolling shutter effect causes the object to appear skewed in the image when the object is moving at high speed. Despite the image skewing drawback, by switching the transmitter faster than the frame rate of the rolling shutter camera, a data rate higher than the frame rate can be achieved due to the row-by-row exposure mechanism. An example is shown in Figure 8c. If a traffic light embedded an array of LEDs blinks at the same speed as the readout rate of each row of the image sensor in this way, the image sensor can receive data at high speed by decoding each row of the image sensor. Each row appears as an independent pattern in the horizontal direction.

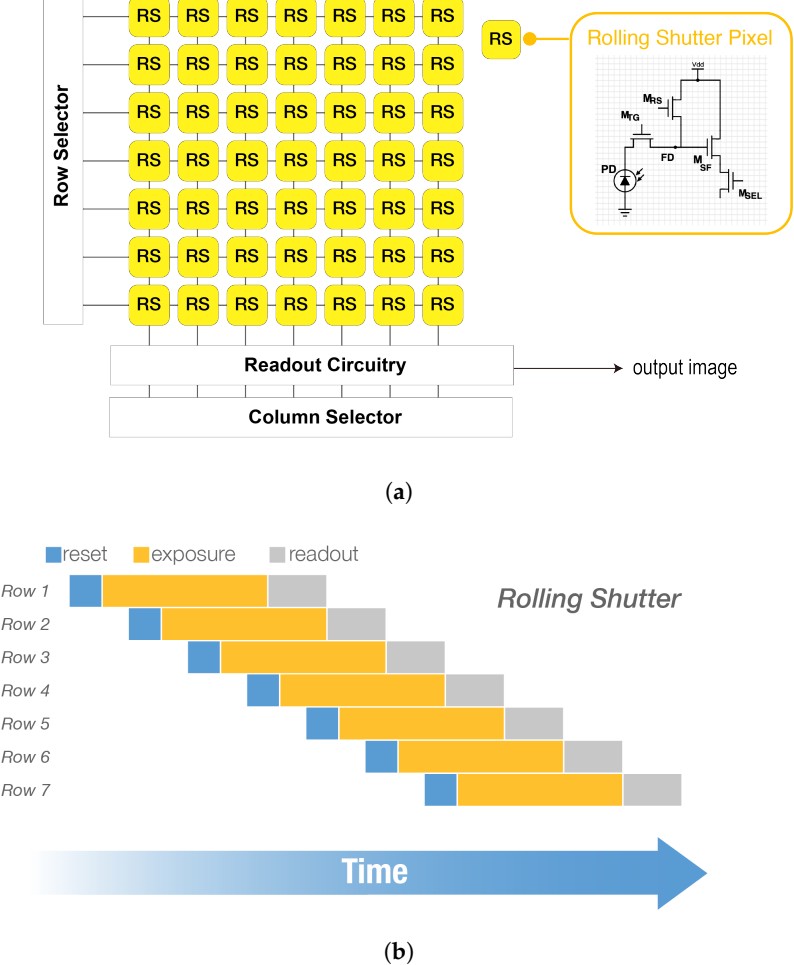

(a)

(b)

**Figure 8.** *Cont.*

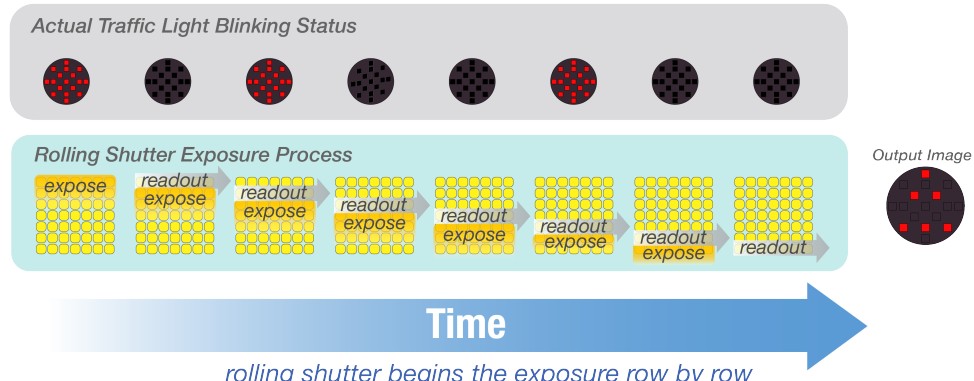

**(c)**

**Figure 8.** The circuit (**a**), exposure mechanism (**b**), and an example (**c**) of a rolling shutter image sensor. (**a**) A simplified rolling shutter image sensor circuit with $7 \times 7$ pixels. The image on the top right shows the basic circuit for a traditional rolling shutter pixel which is a four-transistor active pixel sensor (4T-APS) [46]. A 4T-APS contains a photodiode (PD), a floating diffusion (FD), and four transistors, $M_{RS}$, $M_{TG}$, $M_{SF}$, and $M_{SEL}$. (**b**) The rolling shutter exposure schematic shown over time. The rolling shutter method exposes rows sequentially, followed by an immediate readout. The specific mechanism is decided by the pixel circuit. (**c**) An example of a $7 \times 7$ rolling shutter image sensor capturing a blinking traffic light. We assume that the LED switching rate is equal to the readout rate. The reset is included in the exposure.

In ISC systems that employ rolling shutter effects, the LED blinking rate is higher than the frame rate. Consequently, the images capture stripes due to the LED blinking pattern, such as shown in Figure 8c. In order to decode the stripe image, it is necessary to first extract the LED area in the image, as it is often present in a complex background. Once the LED area has been identified, the stripes can be decoded based on their modulation methods.

The mobility effect of rolling shutter cameras has been investigated in [51,52]. For rolling shutter cameras, which capture the image line-by-line, the motion of the camera during the exposure time can cause distortion or blur in the captured image. This effect can be a significant challenge in ISC systems, where high-quality image capture is essential for reliable communication.

Typically, region-of-interest (RoI) detection methods are used for LED detection. RoI detection is commonly used in computer vision and we can use algorithms such as image thresholding [53] or cam-shift tracking algorithm [54] to extract the LED transmitters. S. Kamiya, et al. have succeeded in error-free detection and communication using a thresholding detecting method when the vehicle is moving at a speed of 15 km/h [55]. Moreover, there are studies conducted to track LEDs by deep learning [52,56].

After completing the LED detection process, we can proceed to the demodulation process. However, the choice of modulation method can significantly impact system performance. In most works the modulation method of OOK is utilized. Other conventional modulation methods include OFDM [52], PWM [57], color-shift keying [58], etc. Furthermore, there are under-sampled modulation methods, including under-sampled frequency shift OOK [59], under-sampled phase shift OOK, spatial-2 phase shift keying (S2-PSK) [60], spatial multiplexing [61], etc.

*4.2. Global-Shutter High-Speed Camera*

Another type of CMOS image sensor used in ISC is the global shutter image sensor [19,62,63]. Global shutter functions differently from the rolling shutter sensors. Instead of exposing rows sequentially, the global shutter exposes each row simultaneously, making it ideal for capturing high-speed motion without skewing. Cameras that use global shutters to capture high-speed motions are called high-speed cameras, and their frame

rates can achieve thousands of fps. According to Section 2.2, 13.8% of the ISC publications used high-speed cameras, which is lower than that of rolling shutter cameras. The main reason is that high-speed cameras are much more expensive than rolling shutter cameras.

Figure 9 provides detailed descriptions of how a global shutter camera works for ISC. Figure 9a shows a conventional five-transistor global shutter sensor circuit. Its scanning mechanism is the same as the rolling shutter that reads out the data row by row. However, the pixel in the global shutter sensor has a memory node, so it can store the accumulated charges. Therefore, the readout process is not required to be conducted immediately after the exposure. In other words, all rows can be exposed simultaneously, and wait a period of time before reading out the charges, as shown in Figure 9b. An example of a global shutter camera capturing a traffic light is shown in Figure 9c. Assume that all the LEDs in the traffic light blink at the same speed as the readout rate of each row of the image sensor, in this way, the image sensor will receive an image of a traffic light with all LEDs illuminated. The output image is different from that of the rolling shutter, even the LED light blinking status is the same. In practice, the LED and the image sensor are usually asynchronous, so the switching speed of the LED must be equal to or less than half of the camera frame rate, according to the Nyquist criterion. As the global shutter high-speed camera captures generally at over 1000 Hz and the data rate is at least half of the frame rate, the data rate can exceed 500 bps.

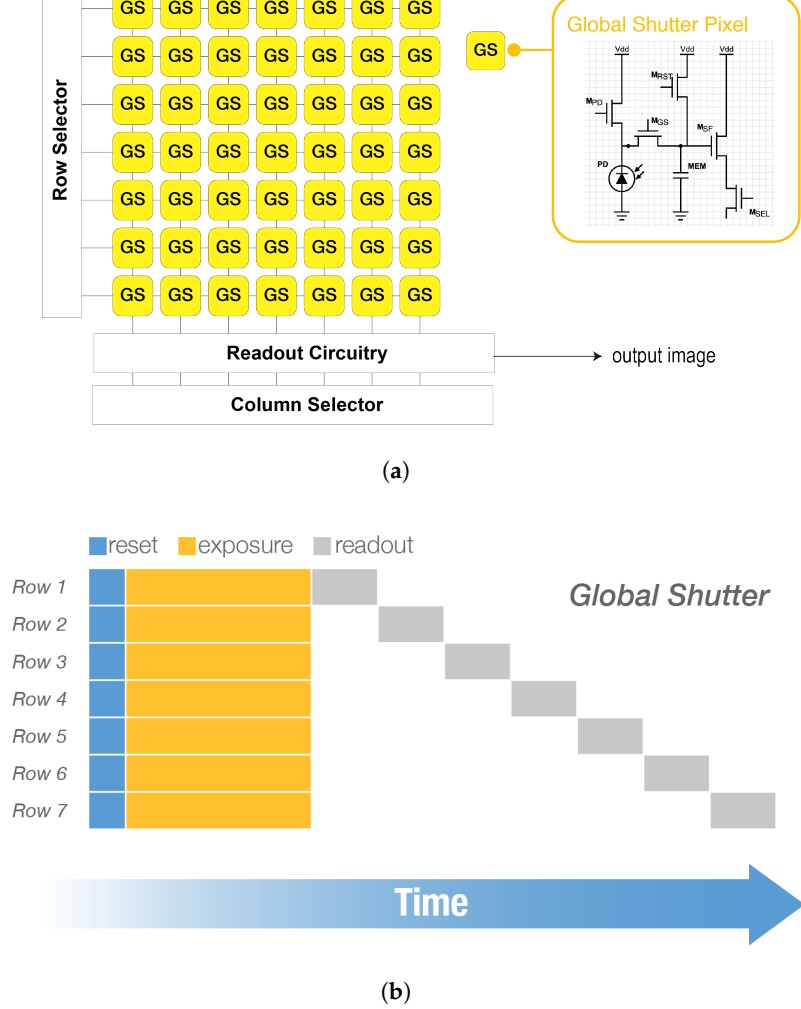

(a)

(b)

**Figure 9.** *Cont.*

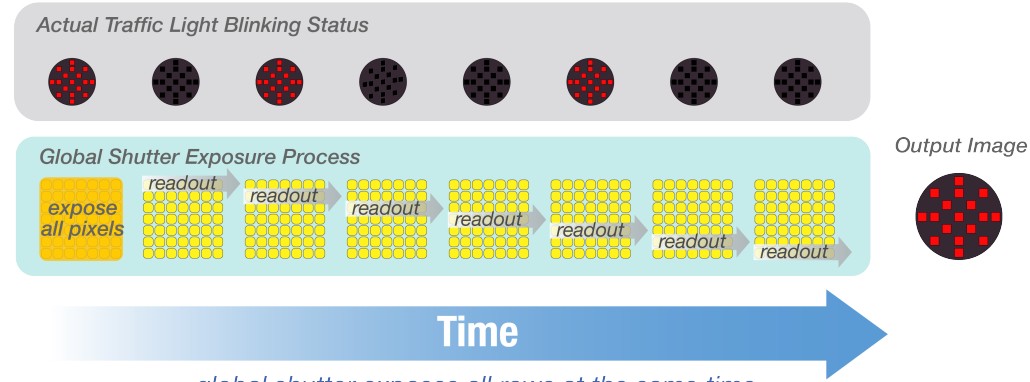

(**c**)

**Figure 9.** A simplified sensor circuit (**a**), exposure mechanism (**b**), and an example of capturing a blinking traffic light (**c**) with a global shutter camera. (**a**) A simplified global shutter image sensor circuit with $7 \times 7$ pixels. The image on the upper-right shows the basic circuit of a traditional five-transistor global shutter active pixel sensor (5T-global shutter APS) [46]. A 5T-global shutter APS contains a photodiode (PD), a memory (MEM) node, and five transistors, $M_{PD}$, $M_{GS}$, $M_{RST}$, $M_{SF}$, and $M_{SEL}$ [46]. The main difference to the rolling shutter pixel is that a global shutter pixel has a memory node for keeping the charge. (**b**) The global shutter exposure schematic shown over time. Global shutter first exposes all the rows simultaneously, and then each row waits for the readout. (**c**) An example of a $7 \times 7$ global shutter image sensor capturing a blinking traffic light. We assume that the LED switching rate is equal to the readout rate. The global shutter only captures the first blinking pattern of the traffic light.

Since the frame rate of a high-speed camera is over 1000 fps, the position of the transmitter on the image differs only slightly from the last consecutive frame. For example, suppose a vehicle has a speed of 50 km/h and the camera has a frame rate of 1000 fps. Then the vehicle moves only 0.01 m in the time period of each frame, and this distance is usually reflected within one pixel on the image plane. As a result, we can take advantage of this feature that high-speed cameras have less displacement in the position of adjacent frames for vehicle tracking, which rolling shutter cameras cannot perform. In [64], T. Nagura et al. proposed an LED array tracking method using inverted signals. The method involves transmitting an inverted pattern immediately after the original LED array pattern, where the entire LED array is obtained by adding these two consecutive patterns. Since the displacement of the LEDs is very small between the two consecutive frames, this method can detect the position of the LED array in the images pretty accurately. However, the data rate is reduced by half. In [65], S. Usui et al. proposed an LED array detection method using spatial and temporal gradients. Spatial refers to the horizontal and vertical gradients of the image of the current frame for which LEDs need to be detected, calculated using Sobel operator; temporal refers to the gradients of the current frame with respect to the previous and next frames, also calculated using Sobel operator. This method identified LED arrays with low spatial-gradient values and high temporal-gradient values. The experiment results showed that error-free LED tracking was achieved when the vehicle was driving at 30 km/h. In [65], the direction of vehicle motion was perpendicular to the LED array plane, while it could also be used when the direction of vehicle motion was parallel to the LED plane [66].

The mobility effect of the vehicle motion has less influence on high-speed image sensors, as shown in [31]. A pinhole camera model was introduced to project world coordinates to image coordinates in [31]. Three types of ISC systems: I2V-ISC, V2I-ISC, and V2V-ISC, were discussed. For I2V-ISC, the camera moved with the vehicle, and the transmitter was static, while for V2I-ISC, the camera was static, and the transmitter moved

with the vehicle. Ref. [31] also compared the vehicle motion models of I2V-ISC and V2I-ISC, and the effects of camera posture on these models. Additionally, Ref. [31] discussed how the relative distance between the transmitter and receiver affects the apparent size and position of objects in the image.

Although the high-speed camera can capture fast motions, it is easily subject to the effect of distance. For a rolling shutter camera, the capturing pattern will not change according to the distance between the vehicle and the transmitter. However, for a global shutter high-speed camera, the size of the transmitter is changing according to the communication distances. Of course, the size of the transmitter on the image is also determined by the image sensor resolution. Hierarchical coding scheme has been proposed in [67] to solve the problem caused by signal degrading at long distance. It divides the LED patterns into different priorities according to the communication distance and assigns different frequencies to their respective priorities by wavelet transform. However, the hierarchical coding scheme has limitations on the number and arrangement of LEDs, which may not match the design of practical transmitters such as traffic lights. To overcome this issue, S. Nishimoto et al. proposed a method of overlay coding in [68,69]. The overlay coding is a more flexible way to design LED applications depending on the transmitter. It distributes long-range or short-range data into large-scale or small-scale of the LED array and encodes them by overlaying each patterns. The experimental results showed that the error-free communication distance of the overlay coding could reach 70 m.

Other limitations of a high-speed camera receiver are the high cost and low resolutions. Due to the advanced technology and components used in high-speed cameras, they can be quite expensive to manufacture and purchase. This can make them cost-prohibitive for smaller organizations that do not have significant budgets for equipment. Another limitation of high-speed camera receivers is their relatively low resolutions compared to other types of cameras. While they are designed to capture images at incredibly high frame rates, the resulting images may not have the same level of detail or clarity as images captured by other types of cameras. This is because high-speed cameras often use smaller sensors and less advanced optics in order to capture images at such high speeds.

### 4.3. Optical Communication Image Sensor (OCI)

In the previous sections, we discussed two types of image sensors used in ISC: rolling shutter and global shutter. Both of these sensors were originally designed to capture images of the entire scene. However, ISC often requires only pixels that contain transmitters. If these pixels can be extracted and attached to a receiver that receives optical signals at high speed, the data rate will be significantly improved. This kind of receiver is proposed in [19,70–75], and known as the OCI. Compared to conventional CMOS image sensors, OCI contains two different kinds of pixels, image pixels (IPx) and communication pixels (CPx), and arranges them side by side as shown in Figure 10. The IPx is a conventional CMOS pixel structure, similar to that of rolling shutter, with an output rate of a few tens of fps. On the other hand, the CPx reduces the capacitance so that it can output the communication signal faster with an output rate of approximately a thousand times that of the IPx [72]. The OCI detects an LED at the IPx and outputs the corresponding CPx in that area. The IPx enables tracking even in a moving vehicle condition. The CPx output is analog, functioning similarly to a PD.

In [72], I. Takai et al. presented the design, fabrication, and capabilities of the OCI, and the experiment results had a 20 Mbps per pixel data rate without LED detection and a 15 Mbps per pixel data rate with real-time LED detection. In [73], I. Takai et al. discussed the capabilities of the OCI used in a V2V communication system, and successfully transmitted vehicle internal data and 13-fps front-view image data of the lead vehicle. In [75], Y. Goto et al. designed optical OFDM into OCI vehicular system, considering frequency response characteristics and circuit noise of the OCI. The system had a performance of 45 Mb/s without bit errors and 55 Mb/s with a BER of $10^{-5}$.

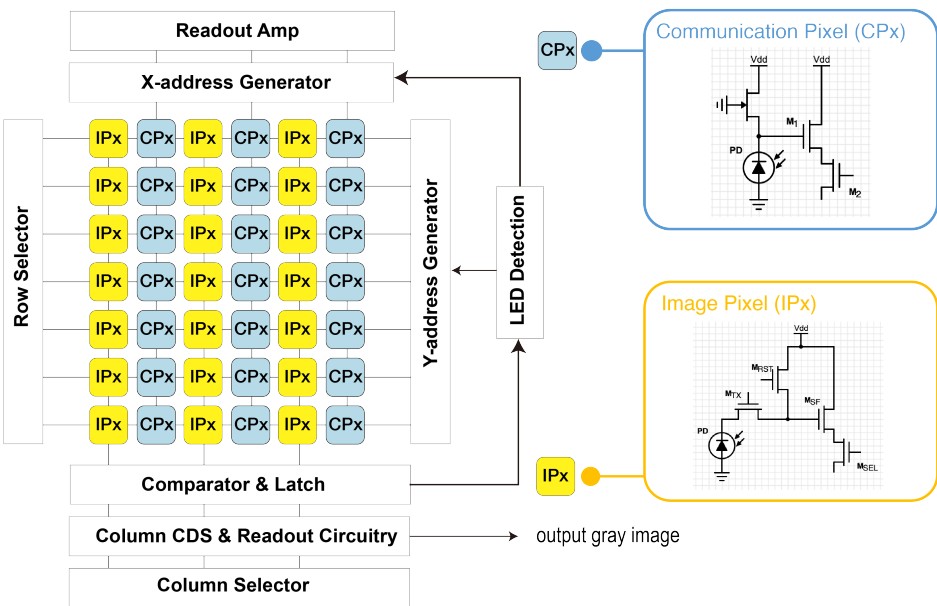

**Figure 10.** A simplified circuit diagram for an optical communication image sensor with 3 × 7 image pixels (IPx) and 3 × 7 communication pixels (CPx). The CPx and IPx are arranged side by side. The two figures on the right show the structure of an IPx and a CPx, adapted from [72]. CDS refers to correlated double sampling, M1 is a transistor used for readout amplifier, M2 is a transistor used for CPx selection, VDD stands for power supply voltage, and Vcs stands for charge-sensing node. Compared to an IPx, the CPx reduce two transistors, making the response faster. Details can be found in [72].

Apparently, the advantage of OCI is the high data rates. It is difficult for conventional CMOS image sensors to achieve data rate at Mbps level. However, the disadvantage of OCI is its low resolution. The current maximum resolution of OCI is 642 × 480 pixels [75], which is much lower than nowadays image sensors. The low resolution may limit its application in long-range communication.

### 4.4. Event Camera (Dynamic Vision Sensor)

Event cameras, also known as dynamic vision sensors or neuromorphic vision sensors, can be used as a receiver of ISC as well. Unlike rolling shutter or global shutter cameras, an event camera does not acquire images with a shutter or frame, but rather record changes in luminance in a pixel and outputs them as asynchronous events [76]. Figure 11 shows the luminance change of a pixel in an event camera and the event output. "ON" refers to positive luminance change and "OFF" refers to negative luminance change.

Since an event camera only outputs positive or negative values, it is simpler to process than a conventional CMOS image sensor, which gives it microsecond-level temporal resolution. Likewise, LEDs can blink at microsecond-level frequencies, so there is an opportunity to achieve ultra-high data rate in ISC with an event camera. In addition, the dynamic range of the event camera is higher than any other conventional CMOS cameras. Its high dynamic range allows the event camera to capture subtle changes in brightness and improve decoding accuracy.

The existing works on the use of event cameras in ISC have only reached a preliminary level. In [77], W. Shen et al. proposed a pulse waveform for event camera-based ISC systems. The experiment compared the inverse pulse position modulation waveform to the proposed pulse waveform, and the communication distance was 3 m and 8 m, respectively. In [78], G. Chen et al. proposed a positioning method using an event camera as a receiver. The proposed system achieved a positioning accuracy of 3 cm when the height between LEDs and the event camera was within 1 m. The low latency and microsecond-

level temporal resolution of the event camera made it possible to identify multiple high-frequency flickering LEDs simultaneously without traditional image processing methods. In [79], Z. Tang et al. investigated a new communication scheme for event camera-based ISC systems using a propeller-type rotary LED transmitter and showed the potential of event-based ISC in moving environment.

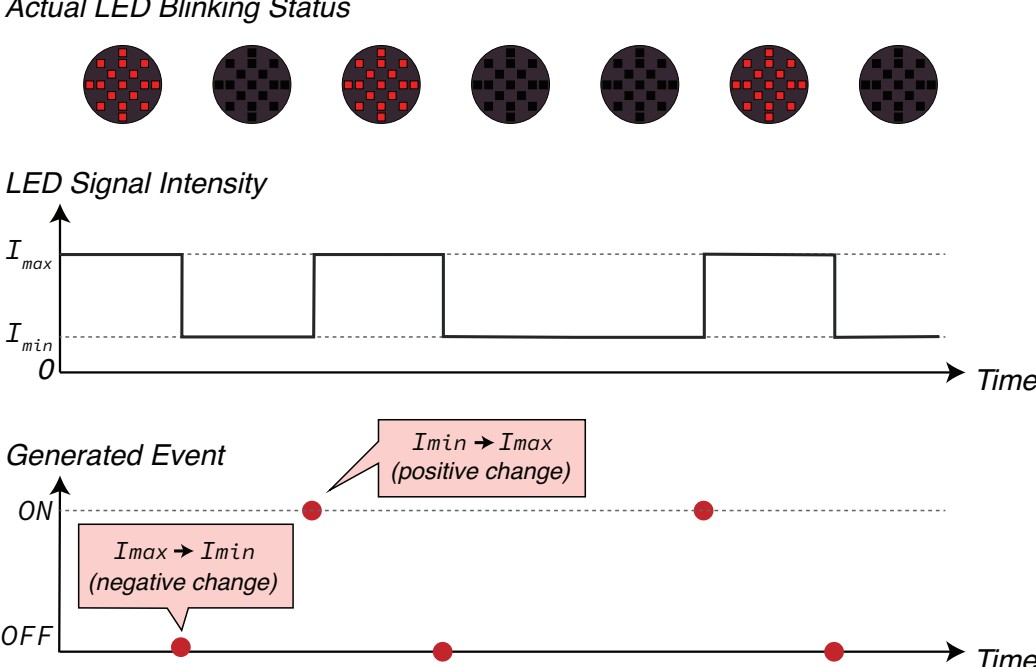

**Figure 11.** Analyzing generated events by correlating with actual LED blinking status and their signal intensity. The $I_{max}$ intensity indicates when the LED is illuminated, while the $I_{min}$ intensity indicates when the LED is off. When a negative luminance change is detected, the event camera generates an "OFF" event, and when a positive luminance change is detected, the event camera generates an "ON" event.

These above works demonstrated the potential of event camera-based ISC systems for low-latency data communication. However, the ability to quickly detect motion also leads to problems with the use of event cameras for ISC. Since event cameras work by detecting changes in luminance, they may have difficulty distinguishing between events generated by motion of LEDs and events generated by changes in luminance. Event cameras may encounter difficulties when the camera or LED is moving because the motion causes changes in luminance, which may be interpreted as LED events, especially if the LED is flashing rapidly or in complex patterns. In addition, as with any imaging system, the event camera is subject to a variety of noise sources, including electronic noise from the sensor and shot noise from the light source. This can make it more difficult to distinguish a true LED event from background noise or other sources of interference. Overall, while event cameras offer some promising advantages for VLC systems, there are still some technical challenges that need to be addressed to realize their full potential.

*4.5. Summary of ISC Receivers*

Table 1 summarizes the ISC receivers discussed in this chapter: rolling shutter camera, high-speed camera, OCI, and event camera. Considering the practical application scenario is important when choosing the appropriate receiver. However, sensor fusion is a possible option that can be applied in many situations.

**Table 1.** A summary of communication performance in existing literature categorizing by image sensor communication (ISC) receivers.

| Receiver Type | Reference | Data Rate | Communication Distance | Vehicle Speed |
|---|---|---|---|---|
| Rolling shutter | [55] | 600∼1000 bps | 5∼70 m | 15∼20 km/h |
| | [61] | 720 bps | 100 m | N/A |
| High-speed camera (global shutter) | [19] | 32 kbps | 30∼65 m | 30 km/h |
| | [43] | 128 kbps | 10∼120 m | N/A |
| | [69] | 40 kbps | 20∼70 m | N/A |
| Optical communication image sensor (OCI) | [19] | 10 Mbps | 20 m | 25 km/h |
| | [72] | 20 Mbps | N/A | N/A |
| | [75] | 55 Mbps | 1.5 m | N/A |
| Event camera | [77] | 16 kbps | 8 m | N/A |

## 5. Range Estimation Using LEDs and Image Sensors

In an ITS, accurate range estimation between a vehicle and its surrounding objects is crucial for ensuring safety. By enabling positioning, the vehicle can determine the distance to its surroundings and take measures to avoid collisions. In an ITS-ISC, image sensors have the ability to obtain depth information through the process of triangulation. It takes advantage of the spatial separation between the viewpoints to calculate the relative positions of the LEDs in the scene, which can then be used to determine their distances to the camera. In this section, we introduce ISC ranging methods based on stereo and monocular vision, covering their principles, challenges, and solutions.

### 5.1. Stereo Vision-Based Range Estimation

The stereo vision-based ISC ranging scheme uses two camera receivers. The distance between the LEDs and the cameras can be determined according to the three-dimensional geometry. Figure 12 illustrates the principle of obtaining range using two cameras and one LED based on the pinhole camera model. The two cameras are considered identical and prior calibration of the two cameras is necessary. Only a single LED is possible to estimate the range between the transmitter and receiver. The cameras are regarded as onboard cameras, and the LED is assumed to be on the lead vehicle or traffic light.

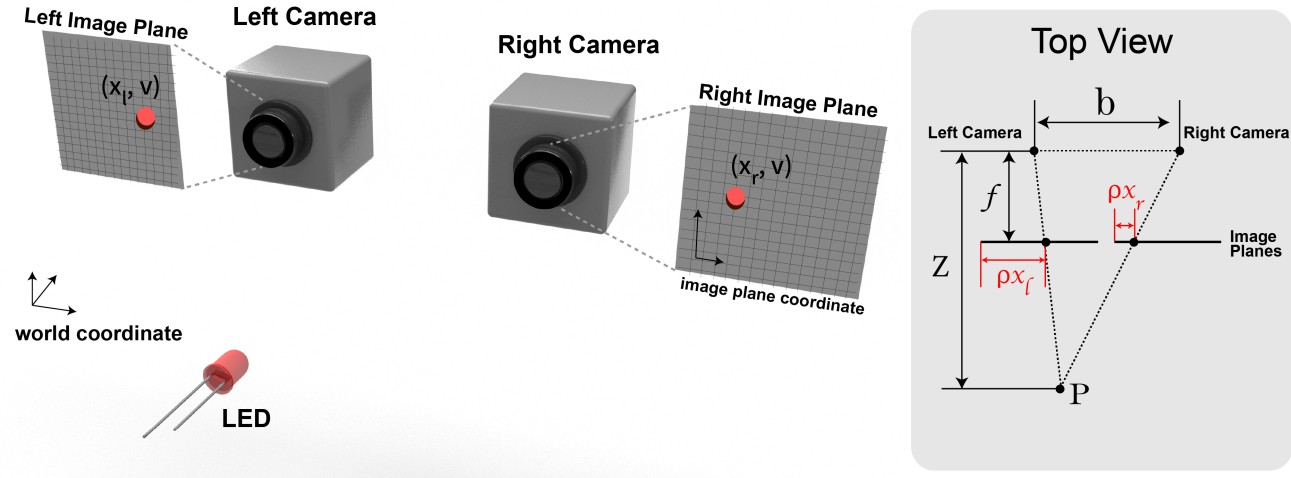

**Figure 12.** A schematic of two cameras capturing a single LED. The two cameras are supposed to be perfectly calibrated. The top view is shown on the right. *P* refers to the LED, *b* is the distance between the left and right camera, *f* is the camera focal length, *ρ* is the size for one pixel, and *Z* is the distance we need to estimate.

The LED is projected into $(x_l, v)$ and $(x_r, v)$ on the image planes of left and right cameras, respectively. According to the triangle similarity criteria, the range $Z$ between the LED and the cameras can be given by

$$Z = f \frac{b}{\rho(x_l - x_r)} \, ,$$

(5)

where $f$ is the focal length of two cameras, $b$ is the distance between the two cameras, and $\rho$ is the actual size per pixel. Thus, if we can find the corresponding feature point on the image plane of the left and right views, the position of the LED can be determined accurately.

The precision of range estimation is significantly affected by the accuracy of computing $(x_l - x_r)$, known as the disparity [80]. In [49], the disparity is calculated using phase-only correlation and the sinc function matching to estimate the disparity in subpixel accuracy. Similarly, Ref. [81] estimated the disparity at the subpixel level using equiangular line fitting, of which the processing speed is faster than phase-only correlation. Furthermore, a positioning algorithm based on neural networks is proposed in [82], and a technique for compensating the rolling shutter effect is proposed in [83].

*5.2. Range Estimation Based on Monocular Vision*

The range between the LED transmitter and the vehicle can also be estimated using a single camera. Figure 13 shows a schematic of using one camera to estimate the distance to the LED.

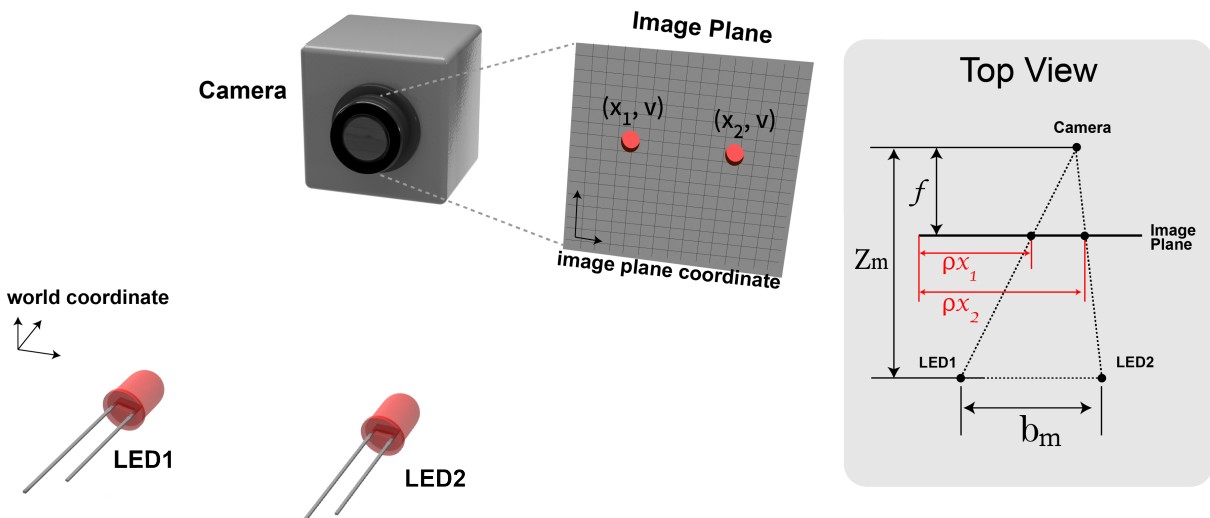

**Figure 13.** The schematic of range estimation using two LEDs and one camera. The top view is shown on the right. $b_m$ is the distance between LED1 and LED2, $f$ is the camera focal length, $\rho$ is the size for one pixel, and $Z_m$ is the distance we need to estimate.

We can use a pinhole camera model to express the transmitter position on both three-dimensional world space and the image plane of the camera. The range from the LED to the camera can be calculated by triangulation, given by

$$Z_m = f \frac{b_m}{\rho(x_2 - x_1)} \, ,$$

(6)

where $Z_m$ is the range between LED and camera, $b_m$ is the distance between LED1 and LED2, $f$ is the focal length of the camera, and $\rho$ is the size per pixel.

However, it is also possible to estimate the range using more than two LEDs based on monocular vision. Reference [84] employed three LEDs to determine the three-dimensional positions of the vehicle, conducting under tunnel scenarios. Reference [85] focused on

the multiple-input and multiple-output (MIMO) cases, where more than two LEDs are required, and utilized the S2-PSK modulation method. Furthermore, when it comes to vehicular applications, it is important to address issues related to vehicle vibration. In [86], a Kalman filter is used to reduce the random errors caused by vehicle moving. In [30], two specialized patterns are introduced to tackle the noise generated by the vibrations of a moving vehicle. Since POC is used for estimating $(x_2 - x_1)$, two LEDs are separately placed on the opposite edge of an LED array in two consecutive frames. The displacement of the upper-left LED on the image plane between the two frames is used to calculate the vehicle vibration. The monocular ranging in [30] is only possible using a high-speed camera, because its time interval must be short enough to compensate a variation caused by moving.

### 5.3. Range Estimation Using Machine Learning

Machine learning or deep learning techniques have been increasingly used for range estimation in vehicular ISC systems [56,82,87]. In [82], a back-propagation (BP) neural-network learning method has been used for positioning and range estimation. In [87], a coding approach is proposed to provide short-distance and long-distance communication. The authors used an artificial neural network (ANN) to forecast the vehicle's location. Long-range communication and high-precision positioning can be conducted simultaneously. The results in [87] showed that the average ranging error was 19.8 mm within a distance of 30 m. Machine learning algorithms can also be combined with traditional signal processing techniques to improve range estimation accuracy. For example, machine learning models can be used to denoise the received signal, compensate for distortion, or enhance the features relevant to range estimation, thereby improving the overall accuracy of the range estimation.

### 5.4. Simultaneous Ranging and Communication

ISC ranging also allows simultaneous communication, making it a more efficient system compared to traditional stand-alone communication or ranging systems. Simultaneous communication saves time, reduces latency, and enhances the overall performance of both communication and ranging. In addition, the hardware complexity of the ISC system can be reduced and the reliability can be improved compared to a separate system because fewer components are required. Furthermore, simultaneous ranging and communication can improve the robustness of the system by reducing the effects of noise and other sources of interference, as well as improving the accuracy of ranging estimations.

Simultaneous ranging and communication can be completed using various approaches. One approach is to use multiple image sensor receivers to receive data signal and estimate the range by stereo vision [56,88]. Another technique is to encode data and ranging information in multiple ISC transmitters, which is then received and decoded by one image sensor receiver [73]. The specific method used depends on the requirements and limitations of the system and the ideal trade-off between communication efficiency and ranging accuracy. Table 2 summarizes the ranging performance in the literature that has been discussed in this review.

**Table 2.** A summary of ranging performance in existing literature.

| Ranging Method | Reference | Ranging Error | Communication Distance | Receiver | Vehicle Speed | Simultaneous Communication |
|---|---|---|---|---|---|---|
| Monocular Ranging | [30] | 0.3 m | 30~60 m | high-speed camera | 30 km/h | No |
| | [73] | N/A | around 8 m | OCI | 12.6~14.0 km/h | Yes |
| | [84] | 1 m | 0~60 m | N/A | N/A | No |
| Stereo Ranging | [88] | 0.5 m | 20~60 m | high-speed camera | N/A | Yes |
| | [83] | 0.1~1.5 m | 0~100 m | rolling shutter camera | 0~100 km/h | Yes |

## 6. Conclusions

This review paper analyzed the research trend and basics of ISC, especially its applications to vehicles. ISC is characterized by the use of a two-dimensional image sensor and, therefore, has spatial separation characteristics. It also has the ability to simultaneously perform communication and ranging. We have made a comprehensive review of various types of ISC receivers (rolling shutter cameras, high-speed cameras, event cameras, and optical communication image sensor) and ISC range estimation techniques. It also highlights the challenges and expected future developments in the field of ISC.

**Author Contributions:** Conceptualization, R.H. and T.Y.; writing—original draft preparation, R.H.; writing—review and editing, T.Y. and R.H.; visualization, R.H.; supervision, T.Y.; project administration, T.Y.; funding acquisition, R.H. All authors have read and agreed to the published version of the manuscript.

**Funding:** This research was funded by the "Nagoya University Interdisciplinary Frontier Fellowship", supported by Nagoya University and JST, the establishment of university fellowships towards the creation of science technology innovation, Grant Number JPMJFS2120.

**Institutional Review Board Statement:** Not applicable.

**Informed Consent Statement:** Not applicable.

**Data Availability Statement:** Not applicable.

**Conflicts of Interest:** The authors declare no conflict of interest.

## Abbreviations

The following abbreviations are used in this manuscript:

| | |
|---|---|
| 4T-APS | four-transistor active pixel sensor |
| BER | Bit-error-rate |
| CCD | Charge-coupled device |
| CMOS | Complementary metal–oxide–semiconductor |
| CPx | Communication pixel |
| DSRC | Dedicated short-range communication |
| ETC | Electronic toll collection |
| FD | floating diffusion |
| fps | frame per second |
| GPS | Global positioning system |
| I2V | Infrastructure to vehicle |
| ISC | Image sensor communication |
| IPx | Image pixel |
| ITS | Intelligent transportation system |
| LED | Light-emitting diode |
| LiFi | Light fidelity |
| LiDAR | Light detection and ranging |
| LOS | Line-of-sight |
| MEM | memory |
| NLOS | Non-line-of-sight |
| OCC | Optical camera communication |
| OCI | Optical communication image sensor |
| OFDM | Orthogonal frequency division multiplexing |
| OOK | On-off keying |
| PD | Photodiode |
| POC | Phase-only correlation |
| S2-PSK | spatial-2 phase shift keying |
| V2V | Vehicle to vehicle |
| V2I | Vehicle to infrastructure |
| V2X | Vehicle to everything |
| VLC | Visible light communication |

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
