# Peer review of "A Review on Image Sensor Communication and Its Applications to Vehicles"

_photonics, doi:10.3390/photonics10060617_

Round 1

Reviewer 1 Report

The authors provide an overview of Image Sensor Communication and its Applications to Vehicles. 

Authors proposed the research trend of VLC and ISC systems.

Beginners may easily understand this manuscript.

Looks interesting  survey paper, but I have some comments for this manuscript:

In section 4, the authors mentioned: ISC receivers include rolling shutter cameras, high-speed cameras, OCI, and event cameras. 

You should include the optical channel and the mobility effect for four image sensors.

In section 5, you mention a range estimation algorithm using stereo vision and monocular vision. 

You should consider applying Machine learning for ranging.

This manuscript is easy to understand for beginners. Still, this paper did not include some future directions and international standardization statuses 

such as IEEE 802.15.7a, ISO OCC, and others about ISC(or OCC). 

Authors may provide some performance comparisons for the IEEE 802.15.7 OWC and IEEE 802.15.7a OCC schemes for V2X applications.

In conclusion, this paper should be updated before publishing.

Reviewer 2 Report

The review article should be accepted for publication after minor revision. The paper is well-written and interesting, and meaningful paper. The results and comparisons are well described. I see the acceptance of the article after making minor comments.

[a]- Add some items and discuss more to RF/free-space optics OWC optical camera communication and effects of channel characteristics and modulation schemes for VLC and the key differences between OCC and other OWC systems

[b]- Deep-Learning Based Methods and Vehicle Detection: Radar-Based Methods.

[c]- References should be improved to be more applicable and practical -- these are some references that may be useful and important for work.

https://doi.org/10.1016/j.optcom.2018.12.034

https://doi.org/10.1515/aot-2020-0038

http://dx.doi.org/10.1109/TITS.2021.3086409

https://doi.org/10.1016/j.optcom.2020.126219

Reviewer 3 Report

The presented manuscript surveys the image sensors photodiodes for VLC. In this sense, it is a useful document for early researchers. I would include the following issues to improve the quality of the paper. 

- Include the aligning issues considering reconfigurable photodetectors, which may include image sensors in vehicles. 

- Effects of full-duplex interference, which plays a major in vehicular applications. 

- Include a remark about the impartance of image sensor for minimising or managing the latency in vehicular communications. 
